# A generic model of life satisfaction: The case study of parkrun

Steve Haake *, Andy Hext, Charlotte Benkowitz

The Advanced Wellbeing Research Centre and the School of Sport and Physical Activity, Sheffield Hallam University, Sheffield, The United Kingdom.

* s.j.haake@shu.ac.uk

## Abstract

Parkrun is a free, weekly, timed 5 km run or walk with the aim of creating 'a healthier, happier planet'. The aim of this study is to use a survey of parkrun to create a model of life satisfaction that can be used by other initiatives seeking (1) to evaluate them using the WELLBY approach, and (2) to create logic models for change in life satisfaction. A cross-sectional survey was sent in autumn 2024–967,478 UK parkrunners producing 78,662 responses. A model was hypothesised and tested using non-linear multiple stepwise regression analysis. Health status was found to have a very large non-linear effect on life satisfaction, primarily related to mental health. Life satisfaction was moderated by age, time registered with parkrun, gender, activity level and index of multiple deprivation. Change in activity level following parkrun participation was found to have a small effect, while the number of runs or walks completed per person had a very small effect. Estimated economic benefits of parkrun to the UK economy were estimated to be £668m. This included £75m for the number of runs or walks completed (£1.92 per run or walk per person), £132m for the increase in activity after participation, and £463m for an estimated improvement to health status of 3% (found in a previous study). The benefit-cost ratio was 53:1 if the estimated increase to health status was included and 16:1 if it was ignored. In terms of health promotion, the model suggested that life satisfaction could be increased most by focussing on sub-populations in the following order: those with very poor, poor and fair health status; those in early middle age; the least active; males; and those from deprived neighbourhoods. Finally, a protocol was described to allow initiatives to create their own simple logic models and their own evaluations.

## Introduction

Research into subjective wellbeing has grown in recent years [1] and governments are increasingly embracing the idea that their role is to improve the wellbeing of the population rather than focus solely on measures such as gross domestic product (GDP) [2,3].

**Data availability statement:** Data used in this study is stored in the Sheffield Hallam University Data Archive at http://doi.org/10.17032/shu-0000000275. Researchers can access the data following the signing of a data sharing agreement.

**Funding:** The survey was funded by parkrun Global (University grant number N940); the principal investigator was AH, the co-investigator was SH. The grant was used to fund CB to carry out the survey. parkrun supported the data collection (as reported in the text) by sending out links for the survey to participants and matching, with consent, surveys to participant data held by them. parkrun had no role in study design, analysis, decision to publish, or preparation of the manuscript.

**Competing interests:** SH was chair of the parkrun Research Board which was closed in 2023. CB did her PhD on parkrun in Australia (2021-2024) which was supported by parkrun through access to participants and their data. AH has no competing interests.

One consequence of this is that the Office of National Statistics in the UK (ONS) has applied four indicators commonly used around the world to try to quantify well-being: life satisfaction, happiness, feeling of things being worthwhile, and anxiety [4]. Life satisfaction is considered to be evaluative, i.e., combines the eudemonic measure of things being worthwhile and the experiential measures of happiness and anxiety, and tends to be the preferred measure of the ONS for the measurement of population wellbeing [5]. On a scale of 0–10, the mean unadjusted life satisfaction in the UK prior to the COVID-19 pandemic was 7.7; this reduced during the pandemic to 7.3 before increasing again by 2024 to 7.5 [4].

Personality traits have been shown to make up at least half the variance in life satisfaction, with health status, economic activity and marital status being strongly associated [6]. Recent ONS data suggests that life satisfaction has a quadratic relationship with both health status and age, reaching a minimum for those between 40 and 49 years of age [7] (see S1 Data). In the UK, females are more likely to report higher life satisfaction scores as are those who live in the least deprived neighbourhoods, although the effects are relatively small [5].

Previous researchers have created models of life satisfaction. A study of 7,954 people from the 1973 National Opinion Research Center Continuous National Survey in the USA found that life satisfaction was associated with age, education, health and marital status [8]. A smaller US study looked at the life satisfaction of 141 older adults: they found that health status and activity were strong predictors of life satisfaction for both males and females, while income influenced life satisfaction indirectly via activity [9]. A study of 245 senior adults in China showed that life satisfaction was positively affected by income, community food provision, social capital and mental health [10], while a structural model of 181 Spanish participants showed that psychological well-being was the largest predictor of life satisfaction [11]. A model of 51 Americans with long term conditions found that social activity was the largest predictor of life satisfaction [12].

The ONS life satisfaction measure is useful as it can also be used to evaluate the economic value of initiatives using the wellbeing adjusted life years (WELLBYs) approach advocated in UK Government guidance [13]. In this, a life satisfaction change of 1 point per person per year in 2019 was worth £13,000 with lower and upper limits of £10,000 and £16,000 [14]. Using this approach, Sport England evaluated the benefits of sport and physical activity to the UK in 2023/4 and showed that it generated over £100 billion per year in social value [13], increasing the WELLBY value using UK Treasury guidance to give a central value of £15,300 in 2023 prices.

One initiative likely to contribute to this Sport England estimate is parkrun, a free, weekly, timed, 5 km run or walk [15], which has 'a healthier and happier planet' as the core of its global strategy [16]. Parkrun has been recommended by the World Health Organization as an initiative that can introduce large numbers of people to the benefits of physical activity [17]. A 2013 study showed that parkrun attracted those less associated with running such as women and those who are least active [18]. A later 2018 cross-sectional study of parkrun showed the health and wellbeing benefits of running or walking [19–22] and volunteering [23]. As well as increases to levels of activity [18], self-reported impacts of parkrun participation have included improvements to fitness,

physical and mental health, and the feeling of being part of a community [18–23]. With these benefits, parkrun has been integrated into the UK's National Health Service as a way of social-prescribing physical activity [24].

A 2019 longitudinal survey of 629 participants showed that seasonally adjusted life satisfaction increased after 6 months by 0.257 on the ONS scale of 0–10 (3.3% of the follow up value) [25]. Change was mediated by mental health, measured using the Short Warwick Edinburgh Mental Wellbeing Score (a scale from 5 to 35) [26], and general health as measured using the EQ-5D visual-analogue score (a scale from 0 to 100) [27]: these increased by 0.8 and 3.0% respectively. Using the wellbeing adjusted life year approach, the authors estimated the cost-benefit ratio to be between 1:17 and 1:99. The wide uncertainty was due to the omission of a direct question that asked participants the impact or 'additionality' of parkrun participation on life satisfaction change, which had to be estimated from proxy data [25].

A new cross-sectional survey of UK parkrunners was conducted in 2024 and had 78,662 survey returns, using a modified version of the 2018 survey [19]. The survey included from the 2018 and 2019 studies, activity level at both registration and the survey (i.e., baseline and follow-up) and the ONS life satisfaction question. The following were added: (1) a health status question used by the ONS; and (2) questions to estimate parkrun's additionality due to running/walking and volunteering.

With the added health status question, the new survey may allow a model to be created to help explain the factors of parkrun (and parkrunners) that most influence life satisfaction. If it is generic enough, it could be used with other initiatives to help create a logic model of their efficacy, and evaluate their cost effectiveness using wellbeing adjusted life years.

### A model of life satisfaction following participation in parkrun: Hypotheses

Previous models suggest the following as hypotheses that life satisfaction is

1. associated with health status [7–11].

2. associated with physical and mental health [7,11].

3. associated, for parkrun, with change in activity level following participation [9,25], and with the number of parkruns completed as a runner, walker or volunteer [18].

4. moderated by demographic variables such as age [7], gender [5], index of multiple deprivation [5], and activity level prior to participation [16,23].

### Aim and objectives of the study

The aim of this study is to create a model of life satisfaction following participation in parkrun: the approach should be transferrable to other initiatives seeking to improve the life satisfaction or wellbeing of a population.

The objectives are as follows:

1. To investigate the impacts following parkrun participation and their association with life satisfaction

2. To create a model of life satisfaction

3. To use the model to evaluate parkrun as a case study

4. To suggest a protocol for other initiatives to allow data collection, evaluation, and the creation of logic models.

## Materials and methods

### Ethics statement

The survey had ethical approval from the ethics committee at Sheffield Hallam University (ER69209902) on 22/8/2024. The survey was sent between 17th and 30th October 2024 to all UK adult parkrunners 18 years and over who had

participated at least once in parkrun in the previous 12 months. Participants were sent a link by parkrun to the survey coded using Qualtrics [28] and were provided with an information sheet to allow them to give informed consent by selecting "I consent to participate in this study". This allowed them to fill out the survey (see S1 Text for full details of wording).

**The survey**

Responses to the following questions drawn from the full survey were used in this study (S1 Text):

1. **Life satisfaction:** "Overall, how satisfied are you with your life nowadays? where 0 is 'not at all satisfied' and 10 is 'completely satisfied'." Respondents were provided with a drop down scale ranging from 0 to 10 [7].

2. **Health status:** "How is your health in general?" Responses allowed were *very poor, poor, fair, good* and *very good.* Responses were coded 0–4 [7].

3. **Activity level at the survey using a question asked at parkrun registration and repeated in the survey**: "Over the last 4 weeks, how often have you done at least 30 minutes of moderate exercise (enough to raise your breathing rate)?" (survey emphasis). Allowed responses were: *less than once per week; about once per week; about twice per week; about three times per week; four or more times per week; rather not say/don't know.* Valid responses were coded 0–4; *rather not say/don't know* were not included.

4. **Gender:** "Please specify your gender." Respondents could answer *male*, *female*, *another gender identity* or *prefer not to say*. Responses were coded 0 for male and 1 for female; other identities were not used.

5. **Impact following participation in parkrun as a runner or walker:** "Thinking about the impact of parkrun, to what extent has running or walking at parkrun changed…" Participants were presented with a randomly ordered list of 19 health and wellbeing outcomes and were asked to rate them using the options *much worse, worse, no impact, better, much better*. Responses were coded 1–5.

6. **Impact following participation in parkrun as a volunteer:** "Thinking about the impact of parkrun, to what extent has volunteering at parkrun changed…" Participants were presented with a randomly ordered list of 20 health and well-being outcomes and were asked to rate them using the options *much worse, worse, no impact, better, much better*. Responses were coded 1–5.

7. **Additionality:** "Thinking about the impact of parkrun, to what extent has running or walking at parkrun changed your life satisfaction" and "Thinking about the impact of parkrun, to what extent has volunteering at parkrun changed your life satisfaction". Responses *much worse, worse, no impact, better, much better* were recoded as -1, -0.5, 0, 0.5 and 1.

The link sent to participants contained an embedded code (SHA-2) which allowed survey data to be matched to the participants' parkrun data. The following matched data from parkrun was used:

1. Age at registration in years (using date of birth)

2. Date of registration (to allow age at the survey to be calculated)

3. Index of multiple deprivation quartile (derived from postcode where Q1 is the most deprived and Q4 the least deprived, coded 1–4)

4. Activity level at registration using the same question used in the survey

5. The total number of parkruns completed

6. The total number of volunteering occasions completed.

The length of time registered was calculated by subtracting the date of registration from the date of the survey. Activity change was calculated by subtracting the coded activity value at registration from the value at the survey to give values between -4 and +4. Due to low numbers with high volunteering participation, above 4 volunteering occasions, 8 categories were used to keep numbers within each broadly consistent. The categories used were 1, 2, 3, 4, 5–6, 7–9, 10–13, 14–19, 20–29, 30–49, 50–99 and >100.

## Cost-effectiveness calculations

The cost-effectiveness of parkrun was estimated using benefit-cost ratios where:

$$Benefit\ cost\ ratio = \frac{Total\ benefit}{Total\ cost}$$

The value of a WELLBY, updated for 2024 prices using Treasury rules was £15,935 with lower and upper limits of £12,257 and £19,612 (see S2 Data).

Two costs were considered in the delivery of parkrun: (1) the cost of running the parkrun organisation; and (2) the cost of park usage (see S3 Data). The cost of running parkrun globally in 2024 was £9.3m [16]. The UK represented around 53% of global registrations such that the pro-rata cost of parkrun for the UK was estimated to be £5.1m. Parkrun delivered 41,300 5 km events in 2024 and it was assumed that this represented 41,300 days of park usage across the UK; this equated to a cost of £7.35m (see S3 Data). Combined together, the 2024 cost of parkrun was £12.5m.

## Statistics

All analysis was carried out using SPSS (IBM: v 26 for Mac). Proportional weights were created using RStudio (2024.12.0) and the Autumn package [29]. These matched the survey distributions to the population of parkrunners who had participated as a runner or walker in the previous 12 months [29]. The following variables were used: age, gender, years registered, number of runs or walks completed, and number of volunteer occasions. Index of multiple deprivation was not included as a weighting variable as this was already similar to the population and remained similar following weighting by other variables (see S4 Data). Cronbach's Alpha was used to assess the reliability of the self-reported items used in the survey, i.e., life satisfaction, health status, activity at registration and at the survey, and impact following running/walking and volunteering.

The following multiple stepwise regressions were performed:

1. A regression for impact following running or walking, and one for impact following volunteering: *your life satisfaction* as the dependent variable; other impacts as linear independent variables; age at the survey, gender, activity at registration and IMD quartile as control variables.

2. A regression for life satisfaction: life satisfaction as the dependent variable; as independent variables, health status (as a quadratic variable), age (as a quadratic variable), index of multiple deprivation quartile, activity at registration, activity change between the survey and registration, the total number of parkruns, and the total number of volunteering occasions.

The effect size for the factors in each multiple regression were estimated using the change in $R^2$ when the factor was added to the regression. The following thresholds suggested by the ONS were used [5]: very small <0.1%; small 0.1 to 0.5%; moderate 0.5 to 1%; large >1%. An additional category was added to the sequence to identify very large as >5%.

Responses for motive and impact were not mandatory such that response counts varied. Approximately 2.6% of participants could not be matched to their parkrun data; their survey responses were included where age, activity at registration, index of multiple deprivation or gender were not required for the analysis.

## Results

### Characteristics of the survey sample

The number of surveys completed was 78,662 with 76,589 (97.4%) matched to parkrun data. When weighted to the population of 969,478 parkrunners, 80.9% were runners or walkers while 19.1% were runners or walkers who also volunteered. The weighted mean years registered was 4.88 years (95% CI ± 0.03 years); the weighted mean number of parkruns completed as a runner or walker was 39 (with a weighted median of 10); the median number of volunteering occasions was 0 (with a mean of 6.4).

Fig 1 shows the following: parkrunners tended to be aged between 16 and 49, with decreasing proportions from 50 to 70+ (Fig 1A); they tended to be from less deprived neighbourhoods (Fig 1B); there were more males than females (52.0 vs 48.0%; Fig 1C); the majority were active three or more days per week at registration (Fig 1D); and the majority had health status at the survey that was good or very good (Fig 1E). The change in activity level between registration and the survey varied between -4 and +4 categories with a mean change of +0.31 categories (Fig1F).

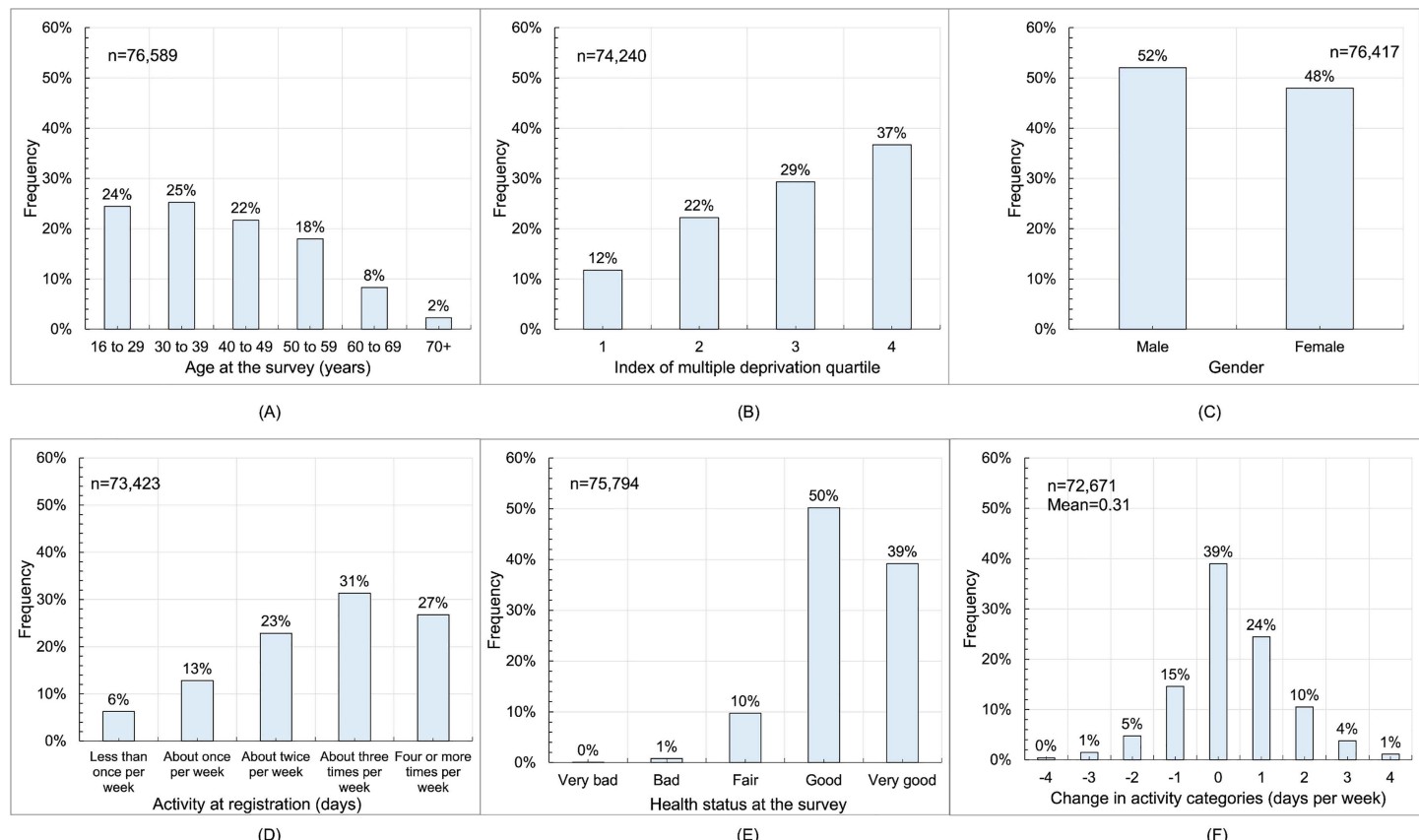

**Fig 1. Demographics of UK parkrunners weighted to those who had participated in the previous year (N = 969,478): (A) age at the survey; (B) index of multiple deprivation quartile (where Q1 is the most deprived and Q4 the least deprived); (C) gender; (D) activity at registration in days per week; (E) health status at the survey; (F) change in activity in categories between registration and the survey.**

**Reliability of survey items**

The reliability of the survey items used in the analysis were assessed using Cronbach's Alpha. The reliability of the scores was found to be 0.935, with a minimum of 0.932 and maximum of 0.941 (see S2 Text). These values were in the range suitable for applied research of 0.8 or more [30].

**Impact following participation in parkrun**

Fig 2A shows the impact measures following participation as a runner or walker. The top three impacts were *your sense of personal achievement* (90%), *your fitness* (88%), and *your physical health* (85%). The bottom three impacts were *the time you have to yourself* (40%), *the number of mew people you meet* (42%) and *your ability to manage your weight* (47%). 76% of participants reported that *your life satisfaction* had been improved through participation as a runner or walker: this made it the 8th ranked impact measure.

Fig 2B shows the impact measures following participation as a volunteer: the top three impacts were *your opportunity to give something back* (93%), *how much you feel part of a community* (84%) and your opportunity to be at parkrun even when not running (81%). The bottom three impacts were *your fitness* (23%), *the time you have to yourself* (23%), and *your physical health* (28%). 73% of participants reported that *your life satisfaction* had been improved through participation as a volunteer: this made it the 5th ranked impact.

**Life satisfaction and additionality following participation in parkrun**

Mean life satisfaction was 7.545 (95% CI ± 0.011). On a scale of -1 to +1 (much worse to much better), the mean impact for *your life satisfaction* following participation as a runner or walker, $I_{rw}$, was 0.454 (95% CI ± 0.005); the mean impact for *your life satisfaction* following participation as a volunteer, $I_v$, was 0.439 (95% CI ± 0.011). Overall impact was calculated using the following:

$$Impact = F_{rw} \times I_{rw} + F_{rwv} \times \frac{(I_{rw} + I_v)}{2}$$

(1)

where $F_{rw}$ is the fraction of runners or walkers (81%) and $F_{rwv}$ is the fraction who are runners or walkers who also volunteer (19%).

This gave an overall value for the sample of 0.453. This value was used as an estimate of parkrun's additionality, i.e., that 45.3% of life satisfaction change can be attributable to parkrun following participation as a runner, walker or volunteer.

**Associations between impact on *your life satisfaction* and other wellbeing impacts**

Table 1 shows results for a multiple stepwise linear regression model of impacts following participating in parkrun as a runner or walker with the impact *your life satisfaction* as the dependent variable. Age, gender, index of multiple deprivation quartile and activity level at registration were offered as controls. Associations with very large effects were *your happiness* and *your mental wellbeing*, while associations with large effects were *your sense of personal achievement, your opportunity to have fun, how active you are*, and *how much you feel part of a community*. Age and gender had moderate and small associations, i.e., older people were more likely and females less likely to report improvements.

Table 2 shows the equivalent table of impacts following participation in parkrun as a volunteer. Associations with very large effects were *your happiness* and *your sense of personal achievement*, while associations with large effects were *your mental wellbeing, how much you feel part of a community,* and *your opportunity to have fun*. Age and gender had small and very small associations, i.e., older people were more likely and females less likely to report improvements.

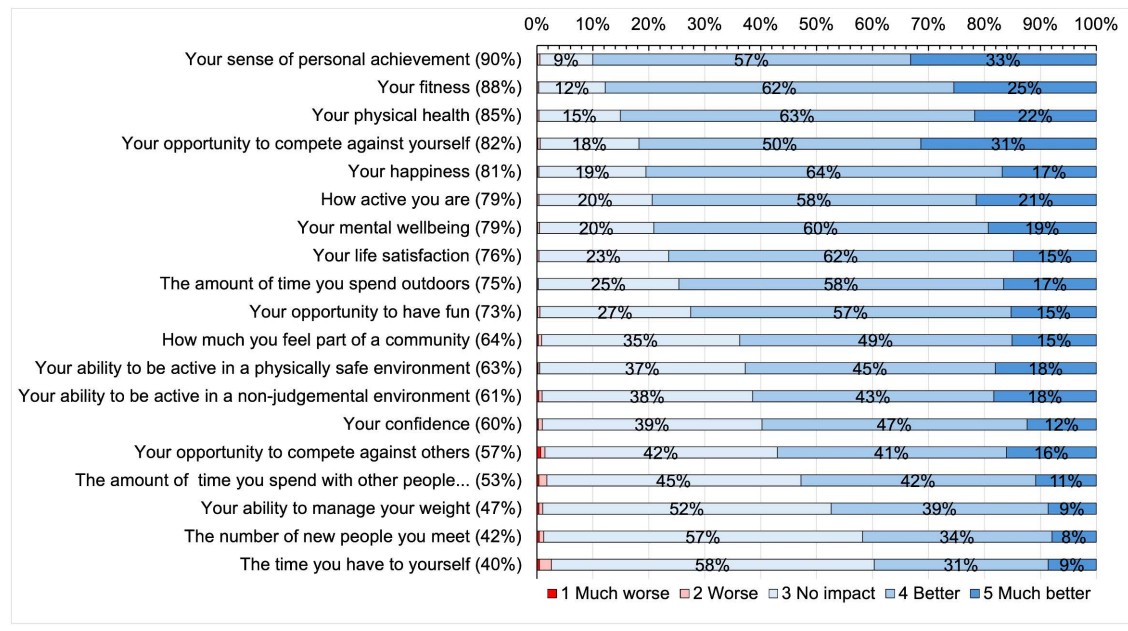

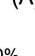

(A)

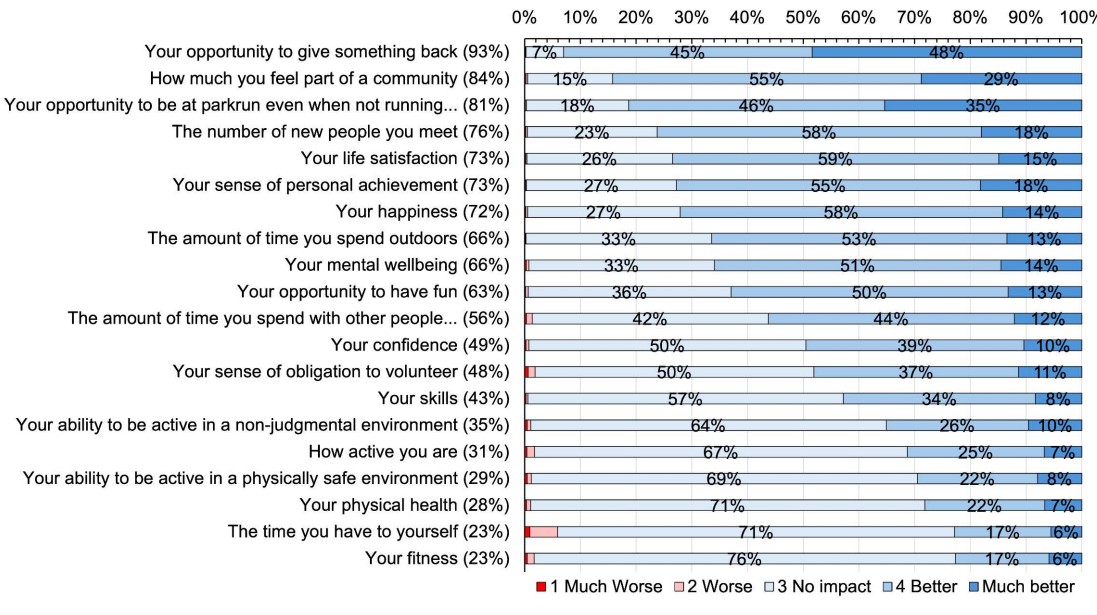

(B)

**Fig 2. Impact following participating in parkrun: (A) "Thinking about the impact of parkrun, to what extent has running or walking at parkrun changed…"; (B) "Thinking about the impact of parkrun, to what extent has volunteering at parkrun changed…"** Percentages in brackets indicate proportions reporting *better* and *much* better.

**Table 1.** Multiple stepwise linear regression model of impact following participation in parkrun as a runner or walker with *your life satisfaction* as the dependent variable. Age, gender, index of multiple deprivation quartile and activity level at registration were offered as controls. Data was weighted to those who had participated in the previous year.

| Factors selected by model | Unstandardised Coefficients | | Standardised Coefficients | | Sig. | R | $R^2$ | $R^2$adj | $R^2$ change | $R^2$ change (% of $R^2$) | Effect[†] |
|---|---|---|---|---|---|---|---|---|---|---|---|
| | B | Standard error | Beta | t | | | | | | | |
| Constant | -0.111 | 0.018 | 0.000 | -6.107 | <0.001 | | | | | | |
| Your happiness | 0.294 | 0.004 | 0.290 | 70.102 | <0.001 | 0.658 | 0.433 | 0.433 | 0.433 | | Very large |
| Your mental wellbeing | 0.142 | 0.004 | 0.148 | 36.915 | <0.001 | 0.693 | 0.480 | 0.480 | 0.0467 | 9.7% | Very large |
| Your sense of personal achievement | 0.078 | 0.004 | 0.080 | 21.058 | <0.001 | 0.708 | 0.502 | 0.502 | 0.0218 | 4.3% | Large |
| Your opportunity to have fun | 0.079 | 0.004 | 0.083 | 22.402 | <0.001 | 0.718 | 0.516 | 0.516 | 0.0145 | 2.8% | Large |
| How active you are | 0.046 | 0.004 | 0.049 | 12.244 | <0.001 | 0.724 | 0.525 | 0.525 | 0.00873 | 1.7% | Large |
| How much you feel part of a community | 0.067 | 0.003 | 0.075 | 20.993 | <0.001 | 0.730 | 0.532 | 0.532 | 0.00752 | 1.4% | Large |
| Your confidence | 0.068 | 0.003 | 0.076 | 20.631 | <0.001 | 0.732 | 0.536 | 0.536 | 0.00401 | 0.7% | Moderate |
| Age at the survey | 0.003 | 0.000 | 0.067 | 22.995 | <0.001 | 0.735 | 0.541 | 0.541 | 0.00442 | 0.8% | Moderate |
| Your physical health | 0.046 | 0.004 | 0.045 | 10.597 | <0.001 | 0.737 | 0.544 | 0.544 | 0.00276 | 0.5% | Moderate |
| The amount of time you spend outdoors | 0.045 | 0.003 | 0.047 | 12.850 | <0.001 | 0.739 | 0.546 | 0.545 | 0.00195 | 0.4% | Small |
| The amount of time you spend with other people (e.g., family, friends or colleagues) | 0.024 | 0.003 | 0.028 | 8.565 | <0.001 | 0.739 | 0.546 | 0.546 | 0.00068 | 0.1% | Small |
| Gender | -0.026 | 0.004 | -0.021 | -7.001 | <0.001 | 0.739 | 0.547 | 0.547 | 0.00062 | 0.1% | Small |
| The time you have to yourself | 0.021 | 0.003 | 0.023 | 7.253 | <0.001 | 0.740 | 0.547 | 0.547 | 0.00058 | 0.1% | Small |
| Your fitness | 0.026 | 0.004 | 0.026 | 6.072 | <0.001 | 0.740 | 0.548 | 0.548 | 0.00043 | 0.08% | Very small |
| Your opportunity to compete against myself | 0.013 | 0.003 | 0.015 | 4.334 | <0.001 | 0.740 | 0.548 | 0.548 | 0.00029 | 0.05% | Very small |
| Your ability to manage your weight | 0.017 | 0.003 | 0.018 | 5.526 | <0.001 | 0.741 | 0.548 | 0.548 | 0.00025 | 0.05% | Very small |
| The number of new people you meet | 0.013 | 0.003 | 0.013 | 3.893 | <0.001 | 0.741 | 0.549 | 0.548 | 0.00013 | 0.02% | Very small |
| Your opportunity to compete against others | 0.009 | 0.003 | 0.011 | 3.499 | <0.001 | 0.741 | 0.549 | 0.548 | 0.00009 | 0.02% | Very small |
| Your ability to be active in a non-judgemental environment | 0.006 | 0.003 | 0.007 | 2.019 | 0.044 | 0.741 | 0.549 | 0.548 | 0.00003 | 0.01% | Very small |
| ANOVA | Sum of squares | df | Mean square | F | Sig. | | | | | | |
| Regression | 12,676 | 19 | 667 | 3,813 | <0.001 | | | | | | |
| Residual | 10,429 | 59,606 | 0.175 | | | | | | | | |
| Total | 23,105 | 59,625 | | | | | | | | | |

[†]Changes in $R^2$: very small<0.1%; 0.1 to 0.5% small; 0.5% to 1% moderate; 1% to 5% large; >5% very large [5].

## A model of life satisfaction

Table 3 shows the multiple non-linear regression models for life satisfaction following participation as a runner, walker or volunteer with $R^2$ change categorised from very large to very small. Health status and health status squared had very large and large effects on life satisfaction; age and age squared had small and very large effects on life satisfaction. This implies that the associate of health and age with life satisfaction is non-linear and that the effect is large or very large.

Gender had a large effect (females 0.157 higher than males), activity at registration a moderate effect (0.055 per category), and activity change between registration and the survey a small effect (0.061 per category). Other factors with small effects were activity at registration (0.047 per category) and deprivation quartile (0.024 per quartile). The time registered

**Table 2.** Multiple stepwise linear regression model of impact following participation in parkrun as a volunteer with *your life satisfaction* as the dependent variable. Age, gender, index of multiple deprivation quartile and activity level at registration were offered as controls. Data was weighted to those who had participated in the previous year.

| Factors selected by model | Unstandardised Coefficients | | Standardised Coefficients | | Sig. | R | R² | R²adj | R² change | R² change (% of R²) | Effect† |
|---|---|---|---|---|---|---|---|---|---|---|---|
| | B | Standard error | Beta | t | | | | | | | |
| Constant | 0.012 | 0.037 | 0.000 | 0.316 | 0.752 | | | | | | |
| Your Happiness | 0.276 | 0.009 | 0.276 | 29.734 | 0.000 | 0.701 | 0.492 | 0.492 | 0.4918 | | Very large |
| Your sense of personal achievement | 0.154 | 0.008 | 0.162 | 19.708 | 0.000 | 0.745 | 0.555 | 0.554 | 0.0627 | 11.3% | Very large |
| Your mental wellbeing | 0.162 | 0.008 | 0.173 | 19.508 | 0.000 | 0.763 | 0.583 | 0.583 | 0.0281 | 4.8% | Large |
| How much you feel part of a community | 0.078 | 0.008 | 0.081 | 9.651 | 0.000 | 0.771 | 0.594 | 0.594 | 0.0116 | 2.0% | Large |
| Your opportunity to have fun | 0.073 | 0.008 | 0.077 | 9.042 | 0.000 | 0.775 | 0.601 | 0.600 | 0.0064 | 1.1% | Large |
| Your opportunity to give something back | 0.070 | 0.007 | 0.069 | 9.478 | 0.000 | 0.777 | 0.604 | 0.604 | 0.0033 | 0.5% | Moderate |
| Your confidence | 0.044 | 0.008 | 0.047 | 5.549 | 0.000 | 0.779 | 0.606 | 0.606 | 0.0021 | 0.3% | Small |
| Age at the survey | 0.002 | 0.000 | 0.041 | 6.954 | 0.000 | 0.780 | 0.608 | 0.608 | 0.0019 | 0.3% | Small |
| The amount of time you spend outdoors | 0.037 | 0.007 | 0.038 | 5.124 | 0.000 | 0.781 | 0.609 | 0.609 | 0.0013 | 0.2% | Small |
| Your skills | 0.024 | 0.008 | 0.024 | 3.078 | 0.002 | 0.781 | 0.610 | 0.610 | 0.0006 | 0.1% | Very small |
| Your sense of obligation to volunteer | 0.015 | 0.005 | 0.017 | 2.782 | 0.005 | 0.781 | 0.610 | 0.610 | 0.0004 | 0.07% | Very small |
| Gender | -0.023 | 0.008 | -0.018 | -3.067 | 0.002 | 0.781 | 0.611 | 0.610 | 0.0003 | 0.05% | Very small |
| The time you have to yourself | 0.016 | 0.007 | 0.017 | 2.445 | 0.015 | 0.782 | 0.611 | 0.610 | 0.0003 | 0.05% | Very small |
| Your ability to be active in a non-judgmental environment | 0.015 | 0.007 | 0.016 | 2.141 | 0.032 | 0.782 | 0.611 | 0.611 | 0.0002 | 0.03% | Very small |
| The number of new people you meet | 0.015 | 0.008 | 0.016 | 1.975 | 0.048 | 0.782 | 0.611 | 0.611 | 0.0001 | 0.02% | Very small |
| ANOVA | Sum of squares | df | Mean square | F | Sig. | | | | | | |
| Regression | 3,018 | 15 | 201 | 1,238 | <0.001 | | | | | | |
| Residual | 1,921 | 11,821 | 0.162 | | | | | | | | |
| Total | 4,939 | 11,836 | | | | | | | | | |

†Changes in R²: very small<0.1%; 0.1 to 0.5% small; 0.5% to 1% moderate; 1% to 5% large; >5% very large [5].

had a large effect (with a very small squared component) as did the number of runs or walks completed per person (0.000266 per run). The number of parkruns completed as a volunteer was not significant.

Fig 3 shows predictions from the model in Table 3 for females active 2 days per week at registration, from a neighbourhood with deprivation score in Q3 (the median for the parkrun population), registered for 4.88 years and who had done 39 parkruns (the weighted means for the parkrun population). The figure shows the non-linear nature of age on life satisfaction which has a minimum at around age 30. Those in very bad health experience the lowest health status with an increase from very bad to bad health increasing life satisfaction by 1.6. In contrast, moving from good to very good health increases life satisfaction by 0.6.

PLOS Global Public Health

**Table 3. Multiple stepwise non-linear regression model of life satisfaction following participation in parkrun as a runner, walker or volunteer.** Dependent variable: life satisfaction change. Independent variables: health status change (as a quadratic), gender, age at registration (as a quadratic), index of multiple deprivation (quartile), activity at registration, activity change, time registered (as a quadratic in years), number of parkruns completed as a runner, walker or as a volunteer.

| Factors selected by model | Unstandardised Coefficients | | Standardised Coefficients | | Sig. | R | $R^2$ | $R^2$adj | $R^2$ change | $R^2$ change (% of $R^2$) | Effect[†] |
|---|---|---|---|---|---|---|---|---|---|---|---|
| | B | Standard error | Beta | t | | | | | | | |
| (Constant) | 3.20 | 0.096 | | 33.15 | <0.001 | | | | | | |
| Health status (categories) | 1.70 | 0.054 | 0.754 | 31.39 | <0.001 | 0.346 | 0.120 | 0.120 | 0.120 | | Very large |
| Age at registration squared (years squared) | 0.000312 | 0.000024 | 0.260 | 13.03 | <0.001 | 0.378 | 0.143 | 0.143 | 0.023 | 16.4% | Very large |
| Health status squared | -0.154 | 0.009 | -0.427 | -17.81 | <0.001 | 0.383 | 0.147 | 0.147 | 0.00399 | 2.7% | Large |
| Gender (male=0; female=1) | 0.157 | 0.011 | 0.0527 | 14.67 | <0.001 | 0.387 | 0.149 | 0.149 | 0.00236 | 1.6% | Large |
| Time registered (years) | 0.0230 | 0.0051 | 0.0561 | 4.49 | <0.001 | 0.389 | 0.151 | 0.151 | 0.00164 | 1.1% | Large |
| Activity change (categories) | 0.0610 | 0.0055 | 0.0515 | 11.15 | <0.001 | 0.389 | 0.152 | 0.152 | 0.00063 | 0.4% | Small |
| Activity at registration (categories) | 0.0554 | 0.0060 | 0.0433 | 9.29 | <0.001 | 0.391 | 0.153 | 0.153 | 0.00111 | 0.7% | Moderate |
| Index of multiple deprivation (quartile) | 0.0336 | 0.0054 | 0.0225 | 6.27 | <0.001 | 0.391 | 0.153 | 0.153 | 0.00048 | 0.3% | Small |
| Age at registration (years) | -0.0112 | 0.0021 | -0.104 | -5.21 | <0.001 | 0.392 | 0.154 | 0.153 | 0.00031 | 0.2% | Small |
| Runs or walks (total) | 0.000266 | 0.00008 | 0.0151 | 3.34 | 0.001 | 0.392 | 0.154 | 0.154 | 0.00012 | 0.08% | Very small |
| Time registered squared (years squared) | -0.000913 | 0.00043 | -0.0270 | -2.15 | 0.032 | 0.392 | 0.154 | 0.154 | 0.00006 | 0.04% | Very small |
| ANOVA | Sum of squares | df | Mean square | F | Sig. | | | | | | |
| Regression | 23,129 | 11 | 2,103 | 1113 | <0.001 | | | | | | |
| Residual | 127,295 | 67373 | 1.889 | | | | | | | | |

[†]Changes in $R^2$: very small<0.1%; 0.1 to 0.5% small; 0.5% to 1% moderate; 1% to 5% large; >5% very large [5].

Fig 3 shows that those in very bad, bad and fair health would have life satisfaction below the UK average of 7.5, while those in very good health would have life satisfaction above it. There is a threshold age for those in good health: below 46 years of age, females would have life satisfaction below the UK average. Increasing activity level by 2 categories increases life satisfaction by about 0.12; decreasing by 2 categories decreases it by about 0.12. This is enough to push a 46 year old female in good health just above or below the UK average.

The logic for the model in Table 3 and Fig 3 is shown diagrammatically in Fig 4. Life satisfaction change is mediated largely by change in health status and, to a lesser extent, by change in activity level and the number of parkruns completed. Life satisfaction is moderated by age, gender, activity at registration and deprivation quartile. It is hypothesised that life satisfaction change is through health status and activity level; these are likely associated with the impacts inside the dashed line.

## Using the model for other survey datasets

The same modelling approach was used to investigate life satisfaction from the previous cross-sectional parkrun survey from 2018 [19]. Rather than the ONS health status question, the EQ-5D VAS question was used to rate general health on a scale of 0–100; otherwise, the same variables were used in the stepwise multiple linear regression, i.e., EQ-5D VAS (as

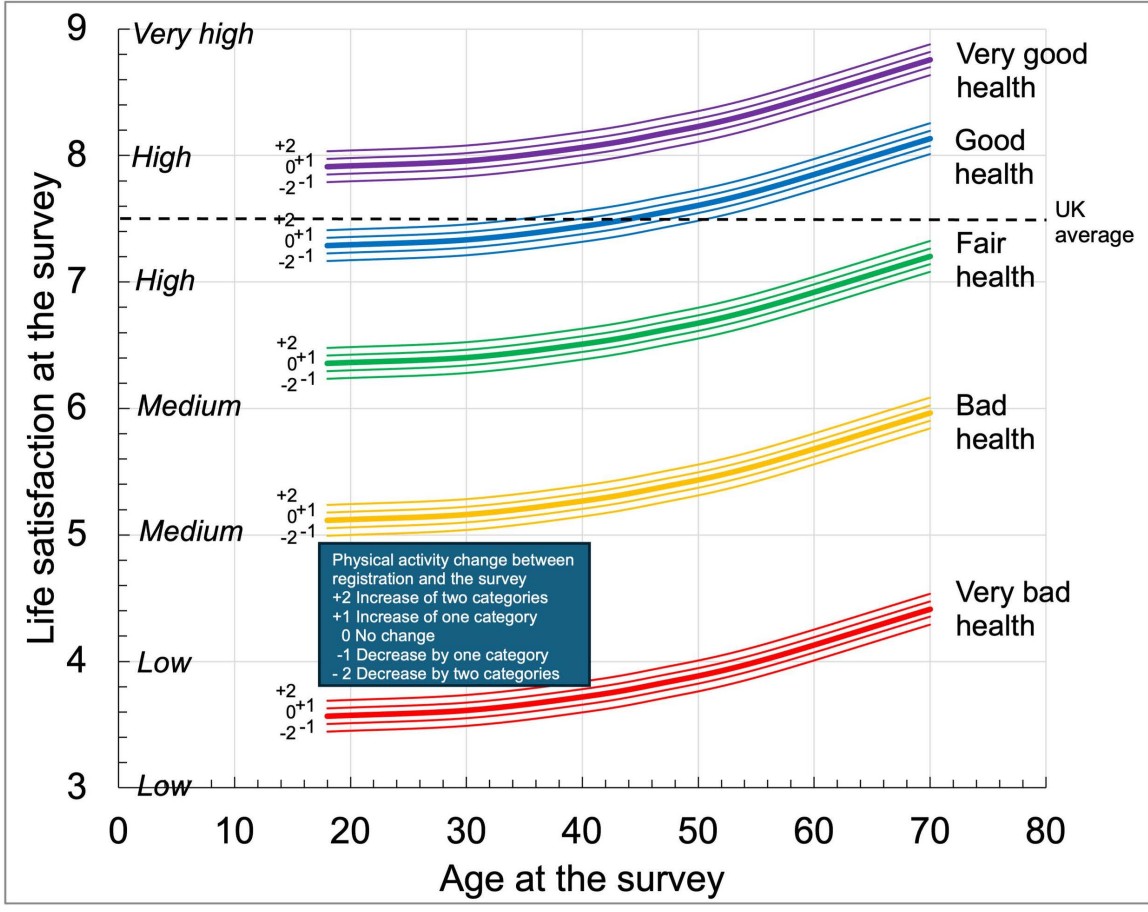

**Fig 3. Multiple regression model for life satisfaction for different health status and activity levels at the survey.** Data shown is for females from neighbourhoods with deprivation score in Q3, active at registration 2 days per week, registered for 4.88 years, and who had completed 39 parkruns as a runner or walker. Office of National Statistics life satisfaction categories also shown [4].

a quadratic), age (as a quadratic), time registered (as a quadratic in years), gender, index of multiple deprivation, activity at registration, number of parkruns completed as a runner or walker and number of volunteering occasions (Fig 5A and Table A, S3 Text). The model for the 2018 cross-sectional survey was similar to that of the 2024 cross-sectional survey: health status had a very large non-linear effect on life satisfaction, while change in activity and the number of parkruns completed as a runner or walker had a small effect. The number of volunteering occasions was not statistically significant. Moderating variables were age (a very large non-linear effect), gender (a small effect), activity at registration (a small effect) and index of multiple deprivation (a small effect).

The same modelling approach was also used on the 2019 longitudinal dataset [25] but because both baseline and follow-up data were available, life satisfaction *change* could be modelled and EQ-5D VAS could be replaced by EQ-5D VAS change. Additionally, the change in Short Warwick Edinburgh Mental Wellbeing Score was included as this had been found to be a mediator in the previous study [25]. Fig 5B (and Table B, S3 Text) shows that change in EQ-5D had a large to very large non-linear effect on life satisfaction change; change in Short Warwick Edinburgh Mental Wellbeing Score also had a very large effect. Change in activity level was found to have a large effect but, neither the number of parkruns

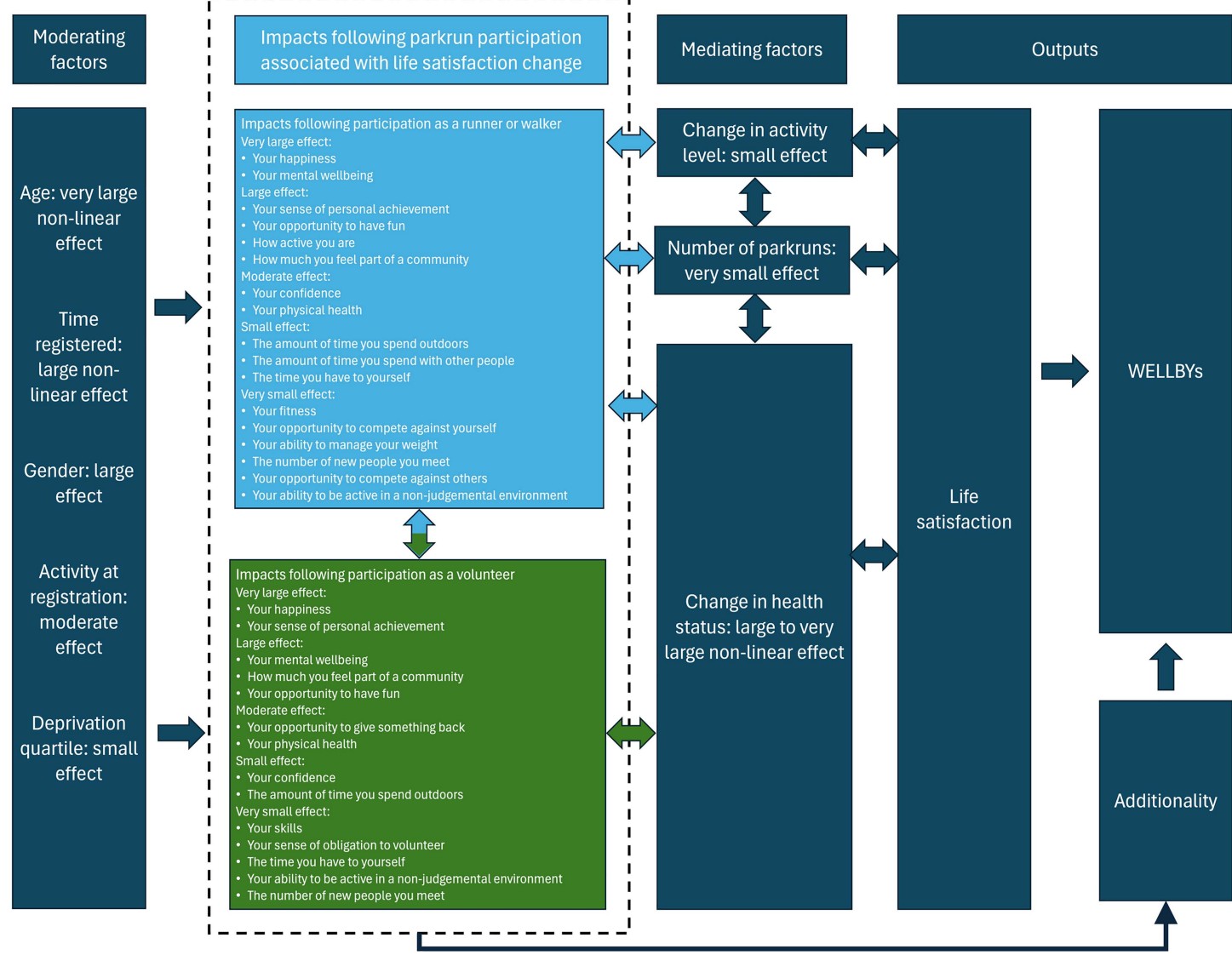

**Fig 4. A model of life satisfaction following participation in parkrun as a runner, walker or volunteer.**

nor the number of volunteering occasions were statistically significant. Moderating variables were age (large non-linear effect) and gender (large effect).

## Using the model to predict change in life satisfaction following participation in parkrun

The model in Table 3 was used to estimate life satisfaction change between registration and the survey due to the number of parkruns completed as a runner or walker, due to physical activity change, and an estimated improvement in health status of 3% as found previously [25] (Table 4). Mean values for the weighted sample were used as a baseline for the model. Table 4 shows that the model predicts a mean life satisfaction of 7.540: this compares to the mean for the weighted sample of 7.545.

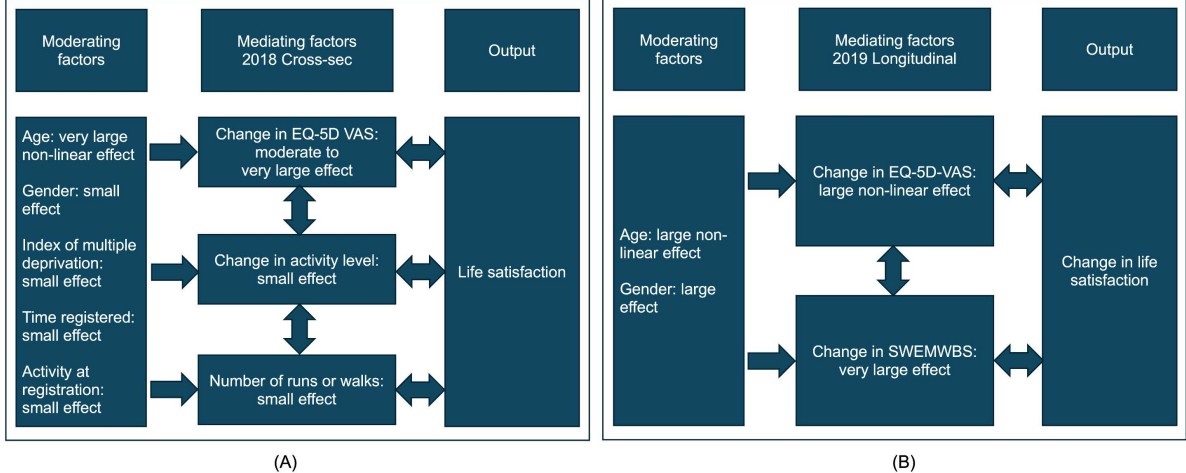

**Fig 5. Comparison of models: (A) life satisfaction from a 2018 cross-sectional study [13]; (B) life satisfaction *change* from a 2019 longitudinal study [19].** See S3 Text for details.

Since the model is a linear regression, individual variables are commutative, i.e., can be added to each other in any order to give the same total. Table 4 shows that completing the mean number of parkruns as a runner or walker of 39 was estimated to give an increase in life satisfaction of 0.0104, while an increase in activity level of 0.31 categories was estimated to give an increase in life satisfaction of 0.0189. Together, these gave a total increase in life satisfaction of 0.0293. An estimated change in health status of 3% gave a change in life satisfaction of 0.0661. The sum for all three variables was 0.0954.

## Estimating parkrun's cost-effectiveness

Additionality of 0.4529 was applied to the life satisfaction estimates in Table 4 to give life satisfaction attributable to parkrun. This was multiplied by the value of a WELLBY (using 2024 prices), with the assumption that participants had taken part in parkrun for at least a year.

The benefit of completing 39 parkruns as a runner or walker was estimated as £75 per person, while the increase in activity of 0.31 was estimated as £137 per person; added together this was £212 per person. For the full 2024 population of 969,478, this equated to £72.7m, £132.4m and £205.0m respectively. The estimated change in health status of 3% was estimated to be £477 per person, with a total value for the full 2024 population of £462.7m. If all three variables were added together, the total value was £667.8m.

The benefit-cost ratio was calculated individually for the number of runs or walks completed, activity change and estimated health status change: these were 5.8 (4.5 to 7.2), 10.6 (8.2 to 13.0) and 37.0 (28.5 to 45.6) to 1. If the number of parkruns and physical activity were considered together, the benefit-cost ratio was 16.4 (12.6 to 20.3) to 1; if all three variables were considered together, the benefit-cost ratio was 53.5 (41.1 to 65.8) to 1.

## Sensitivity analysis of additionality and health status

A sensitivity analysis of additionality and health status change was carried out (Table 5), varying them both between 0 and 100% of the values in Table 4 and comparing the central-value benefit-cost ratios to the value of 5.9:1 found in population studies [31]. With no health status change, i.e., 0%, the benefit-cost ratio was greater than 5.9 if additionality was at least 40% of the original estimate, i.e., 0.181. If additionality was 10% of its original value (i.e., 0.0453), and health status did not change, the benefit cost ratio was 1.6; this increased to 5.3 as health status increased to the full proportion of the 3% estimate of change.

**Table 4. Model estimate of mean life satisfaction with estimates for three variables: (1) the number of parkruns completed as a runner or walker; (2) the change in activity level between registration and the survey; (3) a change in health status of 3%. Data is weighted to those who had participated in the previous year.**

| Factors selected by model | | B | Mean values | 1 Parkruns | 2 Activity change | 3 Health status change (3%) | 1+2 Parkruns & activity change | 1+2+3 Parkruns & activity & health status change (3%) |
|---|---|---|---|---|---|---|---|---|
| (Constant) | | 3.20 | 1 | | | | | |
| Health status (categories) | | 1.70 | 3.277 | | | 0.0987 | | 0.0987 |
| Age at registration squared (years squared) | | 0.000312 | 1,540.6 | | | | | |
| Health status squared (categories squared) | | -0.154 | 11.19 | | | 0.661 | | 0.661 |
| Gender (male=0; female=1) | | 0.157 | 0.480 | | | | | |
| Time registered (years) | | 0.0230 | 4.879 | | | | | |
| Activity change (categories) | | 0.0610 | 0.310 | | 0.310 | | 0.310 | 0.310 |
| Activity at registration (categories) | | 0.0554 | 2.596 | | | | | |
| Index of multiple deprivation (quartile) | | 0.0336 | 2.910 | | | | | |
| Age at registration (years) | | -0.0112 | 36.96 | | | | | |
| Runs or walks (total) | | 0.000266 | 39.04 | 39.04 | | | 39.04 | 39.04 |
| Time registered squared (years squared) | | -0.000913 | 40.99 | | | | | |
| **Life satisfaction (model)** | | | **7.5402** | **0.0104** | **0.0189** | **0.0661** | **0.0293** | **0.0954** |
| Additionality | | 0.4529 | | | | | | |
| **Attributable life satisfaction** | | | | **0.0047** | **0.0086** | **0.0300** | **0.0133** | **0.0432** |
| WELLBY value | Lower | £12,257 | | | | | | |
| | Central | £15,935 | | | | | | |
| | Upper | £19,612 | | | | | | |
| Value per person | Lower | | | £58 | £105 | £367 | £163 | £530 |
| | **Central** | | | **£75** | **£137** | **£477** | **£212** | **£689** |
| | Upper | | | £92 | £168 | £587 | £260 | £848 |
| Number of participants 2024 | | 969,478 | | | | | | |
| Total value 2024 | Lower | | | £55.9m | £101.8m | £355.9m | £157.7m | £513.6m |
| | **Central** | | | **£72.7m** | **£132.4m** | **£462.7m** | **£205.0m** | **£667.8m** |
| | Upper | | | £89.4m | £162.9m | £569.5m | £252.4m | £821.8m |
| Parkrun cost (2024) | | £5.14m | | | | | | |
| Cost of parks (2024) | | £7.35m | | | | | | |
| **Total cost (2024)** | | **£12.5m** | | | | | | |
| Benefit-cost ratio | Lower | | | 4.5 | 8.2 | 28.5 | 12.6 | 41.1 |
| | **Central** | | | **5.8** | **10.6** | **37.0** | **16.4** | **53.5** |
| | Upper | | | 7.2 | 13.0 | 45.6 | 20.2 | 65.8 |

**Table 5. Sensitivity analysis of additionality and health status change for: (a) economic value; and (b) benefit-cost ratios (shading compares to the 5.9:1 for population-level initiatives [31]: see legend). Central WELLBY value of £15,935 used (2024 prices).**

| | | (a) Additionality: proportion of 0.4529 estimate | | | | | | | | | | |
|---|---|---|---|---|---|---|---|---|---|---|---|---|
| | | 0% | 10% | 20% | 30% | 40% | 50% | 60% | 70% | 80% | 90% | 100% |
| | Value | 0.0000 | 0.0453 | 0.0906 | 0.1359 | 0.1812 | 0.2265 | 0.2717 | 0.3170 | 0.3623 | 0.4076 | 0.4529 |
| Health status change: proportion of 3% estimate | 0% 0% | £0.0 | £20.5 | £41.0 | £61.5 | £82.0 | £102.5 | £123.0 | £143.5 | £164.0 | £184.5 | £205.0 |
| | 10% 0.3% | £0.0 | £25.1 | £50.3 | £75.4 | £100.5 | £125.7 | £150.8 | £175.9 | £201.1 | £226.2 | £251.3 |
| | 20% 0.6% | £0.0 | £29.8 | £59.5 | £89.3 | £119.0 | £148.8 | £178.6 | £208.3 | £238.1 | £267.8 | £297.6 |
| | 30% 0.9% | £0.0 | £34.4 | £68.8 | £103.2 | £137.5 | £171.9 | £206.3 | £240.7 | £275.1 | £309.5 | £343.9 |
| | 40% 1.2% | £0.0 | £39.0 | £78.0 | £117.0 | £156.1 | £195.1 | £234.1 | £273.1 | £312.1 | £351.1 | £390.1 |
| | 50% 1.5% | £0.0 | £43.6 | £87.3 | £130.9 | £174.6 | £218.2 | £261.8 | £305.5 | £349.1 | £392.8 | £436.4 |
| | 60% 1.8% | £0.0 | £48.3 | £96.5 | £144.8 | £193.1 | £241.3 | £289.6 | £337.9 | £386.1 | £434.4 | £482.7 |
| | 70% 2.1% | £0.0 | £52.9 | £105.8 | £158.7 | £211.6 | £264.5 | £317.4 | £370.3 | £423.2 | £476.0 | £528.9 |
| | 80% 2.4% | £0.0 | £57.5 | £115.0 | £172.6 | £230.1 | £287.6 | £345.1 | £402.6 | £460.2 | £517.7 | £575.2 |
| | 90% 2.7% | £0.0 | £62.1 | £124.3 | £186.4 | £248.6 | £310.7 | £372.9 | £435.0 | £497.2 | £559.3 | £621.5 |
| | 100% 3% | £0.0 | £66.8 | £133.6 | £200.3 | £267.1 | £333.9 | £400.7 | £467.4 | £534.2 | £601.0 | £667.8 |
| | | (b) Additionality: proportion of 0.4529 estimate | | | | | | | | | | |
| | | 0% | 10% | 20% | 30% | 40% | 50% | 60% | 70% | 80% | 90% | 100% |
| | Value | 0.0000 | 0.0453 | 0.0906 | 0.1359 | 0.1812 | 0.2265 | 0.2717 | 0.3170 | 0.3623 | 0.4076 | 0.4529 |
| Health status change: proportion of 3% estimate | 0% 0% | 0 | 1.6 | 3.3 | 4.9 | 6.6 | 8.2 | 9.9 | 11.5 | 13.1 | 14.8 | 16.4 |
| | 10% 0.3% | 0 | 2.0 | 4.0 | 6.0 | 8.0 | 10.1 | 12.1 | 14.1 | 16.1 | 18.1 | 20.1 |
| | 20% 0.6% | 0 | 2.4 | 4.8 | 7.1 | 9.5 | 11.9 | 14.3 | 16.7 | 19.1 | 21.4 | 23.8 |
| | 30% 0.9% | 0 | 2.8 | 5.5 | 8.3 | 11.0 | 13.8 | 16.5 | 19.3 | 22.0 | 24.8 | 27.5 |
| | 40% 1.2% | 0 | 3.1 | 6.2 | 9.4 | 12.5 | 15.6 | 18.7 | 21.9 | 25.0 | 28.1 | 31.2 |
| | 50% 1.5% | 0 | 3.5 | 7.0 | 10.5 | 14.0 | 17.5 | 21.0 | 24.5 | 28.0 | 31.4 | 34.9 |
| | 60% 1.8% | 0 | 3.9 | 7.7 | 11.6 | 15.5 | 19.3 | 23.2 | 27.1 | 30.9 | 34.8 | 38.6 |
| | 70% 2.1% | 0 | 4.2 | 8.5 | 12.7 | 16.9 | 21.2 | 25.4 | 29.6 | 33.9 | 38.1 | 42.4 |
| | 80% 2.4% | 0 | 4.6 | 9.2 | 13.8 | 18.4 | 23.0 | 27.6 | 32.2 | 36.8 | 41.5 | 46.1 |
| | 90% 2.7% | 0 | 5.0 | 10.0 | 14.9 | 19.9 | 24.9 | 29.9 | 34.8 | 39.8 | 44.8 | 49.8 |
| | 100% 3% | 0 | 5.3 | 10.7 | 16.0 | 21.4 | 26.7 | 32.1 | 37.4 | 42.8 | 48.1 | 53.5 |

**Legend**

| | |
|---|---|
| | Less than 1 |
| (light grey) | Between 1 and 5.9 [31] |
| (dark grey) | Greater than 5.9 |

## Discussion

The model in Fig 4 and the regressions for impact in Tables 1 and 2 reflect previous research showing that hedonistic factors such as happiness and having fun are large or very large components of life satisfaction [7]. This is also true of eudaimonic factors such as a sense of personal achievement. The findings described in Fig 4 and Fig 5 reflect previous models of life satisfaction [7–12] in that life satisfaction is strongly associated with health status [7–9], age [7–8], activity [9], and social activity [12]. One of the main findings here is that mental health is a primary driver of change in life satisfaction, similar to previous findings [10–11].

The non-linear nature of the association between life satisfaction and health status corresponds to findings by the ONS [5,7] and suggests that those in the lowest categories of health status have the greatest capacity for life satisfaction change:

an improvement from very poor to poor health was estimated to increase life satisfaction by 1.6, compared to an increase of 0.6 for those in good health to very good health. This matches the previous longitudinal survey that showed that those with the lowest life satisfaction (and likely the poorest health) increased life satisfaction the most following participation [25].

Two features of parkrun were found to be correlated with life satisfaction: the number of parkruns completed as a runner or walker and the change in activity between registration and the survey. The summation of these effects suggested that for mean values of 39 parkruns and an increase in activity level of 0.31 categories, parkrun participation was associated with a life satisfaction increase of 0.029. An increase in health status of 3% was associated with 2.25 times this value so that the sum of all three values was 0.095. These values are much smaller than the value of 0.25 per person suggested in the previous 6 month longitudinal study [25]. One possible reason for this might be that initial increases in wellbeing may return to previous underlying levels even if the lifestyle change is permanent [32].

Additionality in the study was found to be higher than in the previous study [25] which had a value of 0.17 when both activity change and perceived impact were considered: this is just under 40% of the value found here. Table 5b shows that at this level of additionality, the benefit-cost ratio would be 6.6:1 if there was no change in health status, at least as high as other population initiatives [31].

This study estimates that the benefits of parkruns are £75 per person (£1.92 per run or walk completed per person), the increase in activity is £137 per person, and a 3% health status change is £477 per person, with a total of £689 per person. These benefits are imputed, i.e., they represent the cost to create it if it had been done by other means. For example, a typical cost for an NHS GP appointment would be £45, £145 for an initial mental health service contact, and £345 for a course of psychotherapy [33]. GPs increasingly offer social prescription of services such as art therapy or physical activity (as with parkrun [24]): this costs around £466 per person per year [34]. One might hypothesise that the total mean increase in life satisfaction of around 0.095 per person (0.043 attributed to parkrun) might have reduced some of these potential costs had the participants not taken part in parkrun. Even with the caveats around additionality, the large benefit-cost ratios of at least 6.6:1 (40% of additionality; no health status change) and up to 53:1 show that parkrun is relatively efficient at improving life satisfaction of its participants: this has the potential to mitigate costs in the NHS.

## Strengths and weaknesses of the study

### Strengths

There are a number of strengths of this study. The first is that the principal outcome measure of life satisfaction has been extensively researched by the UK Statistics Authority and has been designated as an accredited official statistic [35]. Likewise, the health status question has been widely used worldwide [36] and a meta-analysis has shown that the ratings correlate inversely with mortality [37].

The Cronbach value for the sample using the survey described here was high at 0.93, greater than the 0.8 cutoff suggested for applied research [30]. This may even be considered too high, suggesting that the questions might have been overly repetitive in asking about subjective wellbeing. While other scales could have been used to measure wellbeing, the life satisfaction is used by governments across many countries worldwide [14] as is the health status question [35]. This makes the approach described here a methodology that might be used in other countries.

Regressions for impact had relatively large $R^2$ values such that 54.8% and 61.1% of the variance in the data was accounted for in the models for running/walking and volunteering, respectively (Tables 1 and 2). In comparison, $R^2$ for life satisfaction was lower with only 14.6% of the variance accounted for; this is similar to ONS regressions for life satisfaction using 14 population variables in which 20.4% of the variance was accounted for [5].

### Weaknesses

The study had two main weaknesses. Firstly, it was cross-sectional and, although activity level was asked at registration and at the survey, the survey was primarily observational. Clearly, a better approach would be to ask both life satisfaction

and health status at both registration and follow-up. Secondly, data was from self-report questionnaires and was likely to contain selection and adherence bias [38]. Since the participants in the study were self-selecting, they represent those who are happy to answer surveys and may represent those with a positive view of parkrun. The weighted mean life satisfaction of the sample was 7.54 while the unweighted value was 7.69 suggests that in terms of demographics at least, the sample represented those with higher levels of life satisfaction at the survey. In terms of adherence, the survey does not represent those who dropped out of parkrun participation or never participated following registration and, again, is likely to represent those with positive views of parkrun.

In terms of the logic model, the self-reported impacts are likely to be overly positive in magnitude. However, given that the regressions in Tables 1 and 2 are relational, the ranking may not change if those with more negative views of parkrun were included in a future survey. It is hypothesised that the ranking of the variables affecting life satisfaction would remain the same even with a less biased sample. Ideally, future research should attempt to find either a matched control group, and perhaps do in-person interviews with those unlikely to fill out online surveys or who have stopped participating in parkrun.

## Implications for other initiatives

The approach described here could be used to evaluate social prescribing and other initiatives seeking to improve the life satisfaction of the population. These might be unrelated to activity but could be initiatives that seek to improve quality of life; examples might include art therapy, music therapy or the use of performing arts [39]. The protocol is broadly as follows:

1. Record at baseline and follow-up, generic measures related to life satisfaction: for example, health status, physical activity etc.

2. Record participant information that might act as moderators of life satisfaction: for example, age, gender, index of multiple deprivation [5].

3. Record life satisfaction at baseline and follow up, with an additionality question such as "Thinking about the impact of , to what extent has it changed your life satisfaction?"

4. Record other initiative-specific information that might be hypothesised to improve life satisfaction at both baseline and follow-up: for example, the number of sessions.

This process should allow the creation of a logic model for the initiative, an estimate of change in life satisfaction of the participants, and its economic value using the wellbeing adjusted life year approach. Care should be taken to understand the causes of life satisfaction change and to use additionality measures to mitigate for attributing change to the correct sources (e.g., initiative or external factors). A knowledge of costs of the initiative would allow a cost-benefit analysis to be carried out.

## Future research

In the development of a generic model, the large number of participants might also be seen as a weakness since other initiatives may have a small number in comparison. Research using this dataset could look at randomly reducing the number of participant responses to investigate the minimum number of participants to allow a valid analysis to be made. Previous research suggests that between 51 [12] and 245 [10] are enough to create complex models. Research should investigate the linearity and validity of the additionality question proposed here and, since parkrun takes place in 23 countries worldwide, research could investigate whether the model created here is valid outside the UK. Research should investigate the attribution of life satisfaction to parkrun or external activities that might be reflected merely by the parkrun population and attempt to assign matched control groups to mitigate for selection and adherence bias. Finally, research could investigate the approach described here with other initiatives unrelated to running or walking.

## Conclusions

A model of change in life satisfaction after participation in parkrun was hypothesised. Life satisfaction was strongly associated with health status (non-linearly), weakly associated with activity change between registration and the survey (a mean time of 4.88 years), and weakly associated with the number of parkruns completed as a runner or walker. Life satisfaction was moderated by age (a large non-linear effect), by gender (a large effect), activity at registration (a moderate effect) and index of multiple deprivation (a small effect). The number of volunteering occasions was not found to be significant. Impacts most strongly associated with life satisfaction change were related to mental health: happiness, mental wellbeing, a sense of personal achievement and the opportunity to have fun. Impacts relating to physical health were less strongly associated with life satisfaction change. In terms of health promotion, the model suggested that life satisfaction could be increased most by focussing on sub-populations in the following order: those with very poor, poor and fair health status; those in early middle age; the least active; males; those from deprived neighbourhoods. The model was used to estimate the impact of parkrun on life satisfaction for UK parkrun participants in 2024 using the wellbeing adjusted life year approach using the following associations: £75m for the number of parkruns completed (£1.92 per run or walk completed per person); £132m for the increase in activity; and £463m for an estimated improvement to health status of 3% (found in a previous study). The total estimated benefits for 2024 were £668m. The benefit-cost ratio was 53:1 if the estimated increase to health status was included and 16:1 if it was ignored. The study contained selection and adherence biases likely leading to overly positive reporting of life satisfaction and impact. A sensitivity analysis to mitigate for this showed that parkrun was likely to be at least as cost effective as other population-level initiatives. The approach described here could be used to create models of life satisfaction for other initiatives seeking to improve health and wellbeing (not necessarily related to physical activity), and to evaluate them.

## Supporting information

**S1 Data. Office of National Statistics data on life satisfaction.**
(XLSX)

**S2 Data. Office of National Statistics inflation figures applied to the value of a WELLBY.**
(XLSX)

**S3 Data. Cost of running parkrun events.**
(XLSX)

**S4 Data. Weighting of sample compared to parkrun population.**
(XLSX)

**S1 Text. 2024 UK survey of parkrun.**
(DOCX)

**S2 Text. Reliability of survey questions.**
(DOCX)

**S3 Text. Models using previous parkrun surveys.**
(DOCX)

## Acknowledgments

Many thanks to Chrissie Wellington and Mike Graney at parkrun for enabling the distribution of the survey, and to the participants who filled it out.

## Author contributions

**Conceptualization:** Steve Haake.

**Data curation:** Steve Haake, Charlotte Benkowitz.

**Formal analysis:** Steve Haake.

**Funding acquisition:** Andy Hext, Charlotte Benkowitz.

**Investigation:** Steve Haake, Charlotte Benkowitz.

**Methodology:** Steve Haake.

**Project administration:** Andy Hext, Charlotte Benkowitz.

**Resources:** Charlotte Benkowitz.

**Supervision:** Steve Haake, Andy Hext.

**Validation:** Charlotte Benkowitz.

**Visualization:** Steve Haake.

**Writing – original draft:** Steve Haake.

**Writing – review & editing:** Steve Haake, Andy Hext, Charlotte Benkowitz.

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
