## [Decision Letter · Decision Letter 0]

20 May 2025

PGPH-D-25-00740

A generic model of life satisfaction: the case study of parkrun

Dear Dr. Haake,

Thank you for submitting your manuscript to PLOS Global Public Health. After careful consideration, we feel that it has merit but does not fully meet PLOS Global Public Health’s publication criteria as it currently stands. Therefore, we invite you to submit a revised version of the manuscript that addresses the points raised during the review process.

EDITOR: Dear Author, please revise the manuscript according to the reviewers' suggestions to enhance its scientific merit.

We look forward to receiving your revised manuscript.

Kind regards,

Zulkarnain Jaafar

Academic Editor

Additional Editor Comments (if provided):

Reviewers' comments:

Reviewer's Responses to Questions

**Comments to the Author**

1. Does this manuscript meet PLOS Global Public Health’s publication criteria ? Is the manuscript technically sound, and do the data support the conclusions? The manuscript must describe methodologically and ethically rigorous research with conclusions that are appropriately drawn based on the data presented.

Reviewer #1: Yes

Reviewer #2: Yes

2. Has the statistical analysis been performed appropriately and rigorously?

Reviewer #1: No

Reviewer #2: Yes

3. Have the authors made all data underlying the findings in their manuscript fully available (please refer to the Data Availability Statement at the start of the manuscript PDF file)?

Reviewer #1: Yes

Reviewer #2: Yes

4. Is the manuscript presented in an intelligible fashion and written in standard English?

Reviewer #1: Yes

Reviewer #2: Yes

5. Review Comments to the Author

Reviewer #1: Please provide reliability tests (i.e. Cronbach alpha scores) of all research instruments (questionnaires) used in this study to obtain data (Life Satisfaction, health status etc.). It will add required credibility to this study. Please critically discuss the reliability results of these instruments and provide relevant justification for using them as opposed to other research instruments

Reviewer #2: This manuscript is an effectively carried out and timely examination of life satisfaction as an indicator of well-being based on extensive national survey data collected from parkrun participants in the UK. The authors construct a theory-driven and methodologically rigorous model for investigating drivers for life satisfaction using rigorous regression methods and cost-effectiveness analyses. Applying the WELLBY framework for the broader measurement of the economic benefits of parkrun participation adds a refreshing and entirely applicable element for public health policymaking and evaluation practice.

The manuscript has a number of admirable strengths. Using a large, weighted dataset increases the internal validity, and the fact that it incorporates health status, activity change, and parkrun participation enables a multi-dimensional understanding of life satisfaction. Additionally, methodological transparency, ranging from ethics to regression diagnostics and sensitivity analysis, enhances the scientific credibility of the study. Attempting to make the model applicable to other health promotion activities using clear protocol is particularly beneficial to practitioners and policymakers.

Then again, a few areas require revision for greater clarity and impact. First, the authors must explain the rationale for applying WELLBY valuations based on 2019 and why adjustments for 2024 economic conditions are unnecessary. Second, while the authors recognize the limitations inherent in cross-sectional data, more discussion on assumed selection biases and whether and how they might affect additionality estimates would enhance the discussion. Third, while parkrun activity is shown not to be significantly related to the frequency of life satisfaction (number of events), the discrepancy between this finding and reported impact (self-reported improvement) must be explored further—possibly accounting for social desirability bias or recall effects. Resolving the former three issues would add more explanatory power and interpretative balance to the manuscript.

6. PLOS authors have the option to publish the peer review history of their article (what does this mean? ). If published, this will include your full peer review and any attached files.

**Do you want your identity to be public for this peer review?** For information about this choice, including consent withdrawal, please see our Privacy Policy .

Reviewer #1: **Yes: ** Dr Adeniyi Abolaji Adeboye

Reviewer #2: **Yes: ** Abimbola Adegoke

---

## [Decision Letter · Decision Letter 1]

30 Jul 2025

A generic model of life satisfaction: the case study of parkrun

PGPH-D-25-00740R1

Dear Dr Haake,

We are pleased to inform you that your manuscript 'A generic model of life satisfaction: the case study of parkrun' has been provisionally accepted for publication in PLOS Global Public Health.

Best regards,

Zulkarnain Jaafar

Academic Editor

Reviewer Comments (if any, and for reference):

Reviewer's Responses to Questions

**Comments to the Author**

1. If the authors have adequately addressed your comments raised in a previous round of review and you feel that this manuscript is now acceptable for publication, you may indicate that here to bypass the “Comments to the Author” section, enter your conflict of interest statement in the “Confidential to Editor” section, and submit your "Accept" recommendation.

Reviewer #1: All comments have been addressed

Reviewer #2: All comments have been addressed

2. Does this manuscript meet PLOS Global Public Health’s publication criteria ? Is the manuscript technically sound, and do the data support the conclusions? The manuscript must describe methodologically and ethically rigorous research with conclusions that are appropriately drawn based on the data presented.

Reviewer #1: Yes

Reviewer #2: Yes

3. Has the statistical analysis been performed appropriately and rigorously?

Reviewer #1: Yes

Reviewer #2: Yes

4. Have the authors made all data underlying the findings in their manuscript fully available (please refer to the Data Availability Statement at the start of the manuscript PDF file)?

Reviewer #1: Yes

Reviewer #2: Yes

5. Is the manuscript presented in an intelligible fashion and written in standard English?

Reviewer #1: Yes

Reviewer #2: Yes

6. Review Comments to the Author

Reviewer #1: All comments have been addressed

Reviewer #2: Thank you so very much for your careful and thoughtful reworking. The paper contributes even better to public health policy literature by marrying theoretical sophistication with empirical salience. Specific strengths are:

Application of Cronbach's alpha (α = 0.93) to determine reliability of survey instruments, well within acceptable levels suitable for applied research.

WELLBY valuation update to 2024 Treasury-adjusted projections, cementing fiscal relevance.

New sensitivity analysis and explicit explanation of selection/adherence bias, which offer suitable caution and enhanced generalizability.

Clarification on attendance frequency at parkrun and life satisfaction, later corrected by regression disaggregation of age influence and attendance history.

Large tables and figures (Table 4, 5, Figures 3–5a) all accompany the account with clarity and rigor.

These advances have substantially increased interpretative capacity, usability, and potential to transform community-based measures into interventions involving wellbeing. I vote publish.

7. PLOS authors have the option to publish the peer review history of their article (what does this mean? ). If published, this will include your full peer review and any attached files.

**Do you want your identity to be public for this peer review?** For information about this choice, including consent withdrawal, please see our Privacy Policy .

Reviewer #1: **Yes: ** Dr Adeniyi Abolaji Adeboye

Reviewer #2: **Yes: ** Abimbola Adegoke
